# Wide Real-Life Data Support Reduced Sensitivity of Antigen Tests for Omicron SARS-CoV-2 Infections

**DOI:** 10.3390/v16050657

**Published:** 2024-04-23

**Authors:** Chiara Piubelli, Davide Treggiari, Denise Lavezzari, Michela Deiana, Klevia Dishnica, Emma Maria Sole Tosato, Cristina Mazzi, Paolo Cattaneo, Antonio Mori, Elena Pomari, Lavinia Nicolini, Martina Leonardi, Francesca Perandin, Fabio Formenti, Alejandro Giorgetti, Antonio Conti, Maria Rosaria Capobianchi, Federico Giovanni Gobbi, Concetta Castilletti

**Affiliations:** 1Department of Infectious, Tropical Diseases and Microbiology, IRCCS Sacro Cuore—Don Calabria Hospital, Negrar di Valpolicella, 37124 Verona, Italylavinia.nicolini@sacrocuore.it (L.N.);; 2Department of Biotechnology, University of Verona, 37124 Verona, Italy; 3Centre for Clinical Research, IRCCS Sacro Cuore—Don Calabria Hospital, Negrar di Valpolicella, 37124 Verona, Italy; cristina.mazzi@sacrocuore.it; 4Clinical Analysis Laboratory and Transfusional Service, IRCCS Sacro Cuore—Don Calabria Hospital, Negrar di Valpolicella, 37124 Verona, Italy

**Keywords:** SARS-CoV-2, diagnosis, antigenic rapid test, molecular test, nucleocapsid protein, Omicron variant, Delta variant, NGS, nose site sampling, mouth site sampling, variant of concern (VOC)

## Abstract

With the continuous spread of new SARS-CoV-2 variants of concern (VOCs), the monitoring of diagnostic test performances is mandatory. We evaluated the changes in antigen diagnostic tests’ (ADTs) accuracy along the Delta to Omicron VOCs transition, exploring the N protein mutations possibly affecting ADT sensitivity and assessing the best sampling site for the diagnosis of Omicron infections. In total, 5175 subjects were enrolled from 1 October 2021 to 15 July 2022. The inclusion criteria were SARS-CoV-2 ADT combined with a same-day RT-PCR swab test. For the sampling site analysis, 61 patients were prospectively recruited during the Omicron period for nasal and oral swab analyses by RT-PCR. Next-Generation Sequencing data were obtained to evaluate the different sublineages. Using RT-PCR as a reference, 387 subjects resulted in becoming infected and the overall sensitivity of the ADT decreased from 63% in the Delta period to 33% in the Omicron period. This decrease was highly statistically significant (*p* < 0.001), and no decrease in viral load was detected at the RNA level. The nasal site presented a significantly higher viral load than the oral site during the Omicron wave. The reduced detection rate of Omicron infections by ADT should be considered in the global testing strategy to preserve accurate diagnoses across the changing SARS-CoV-2 variants.

## 1. Introduction

As of 31 February 2024, over 775 million confirmed cases of the novel coronavirus disease (COVID-19) caused by severe acute respiratory syndrome coronavirus 2 (SARS-CoV-2) and over 7 million deaths have been reported globally [1]. Although on 5 May 2023, the World Health Organization (WHO) declared the end of the COVID-19 global public health emergency, WHO still recommends governments to maintain a monitoring system, in case new variants emerge and cause another surge. Moreover, the accurate identification of people infected with SARS-CoV-2 is an essential prerequisite for facilitating the early initiation of therapy to reduce disease progression and for limiting the community spread of the infection [2].

The appearance and evolution of new variants with novel mutations require the monitoring of the available diagnostic methods for the detection of SARS-CoV-2 infection, based on both molecular and antigen testing. With the emergence of Omicron in particular, the effectiveness of antigen diagnostic tests (ADTs) was questioned. Diagnostic test sensitivity is a major criterion for detecting individuals infected with SARS-CoV-2 as fast as possible [3]. Most commercially available ADTs are based on the detection of the Nucleocapsid (N) protein, one of the four major structural proteins of SARS-CoV-2 [4], which has proven to be a good diagnostic target due to its high conservation rate [5,6,7,8]. However, mutations also affect this gene. In fact, ADTs were developed for the original SARS-CoV-2 N protein, and since the initial phases of the pandemic, new viral variants have been identified with specific patterns of mutations that could affect their detection due to epitope modification.

In October 2021, the Omicron variant (B.1.1.529) emerged in South Africa and started to be the dominant SARS-CoV-2 variant worldwide [9,10,11]. Omicron and its descendent sub-variants drew particular attention due to the high number of mutations. Their higher transmissibility and immune escape ability were assessed compared to the Delta (B.1.617.2) variant [12]. But how alteration in the N protein could influence antigen recognition by diagnostic tests has never been clarified. Omicron sub-variants have extensive mutations in its spike (S) and N proteins [13]. Mutations in the Nucleocapsid gene may lead to protein conformational changes that affect the target binding site of the ADT. This could, theoretically, alter the performance of the ADT in detecting this variant [14,15,16,17]. The rapid global emergence and dominance of the Omicron variant highlighted the importance of understanding the performance of ADTs in real-world settings. Some in vitro studies suggested that the performance of rapid ADTs did not differ between the Delta and Omicron variants [18,19], while studies using clinical specimens suggested a possible decrease in antigen tests’ sensitivity for the Omicron variant [20,21,22,23].

In addition to the variability of the N protein, in the early months of 2022, some studies hypothesized that Omicron variant infection could present a higher level of detectable viral RNA in the mouth than in the nose, with a positive predictive value of 100% in the saliva compared with 86% in mid-turbinate swabs [24]. These findings were supported by data from other labs, describing altered tissue tropism for the Omicron variant [25]. Another study did not support a preferred sample type for Omicron detection, but suggested a heterogeneous distribution of viral RNA in the nose and mouth [26], indicating that the choice of the sampling site still remains a controversial issue.

Based on the above considerations, further studies are needed to monitor the performance of these diagnostic tests in order to maintain accurate diagnoses throughout the evolution of the Omicron variant. Therefore, the aim of our study was to directly compare the results of ADTs with those of corresponding molecular tests in the same subjects, in a cohort of about 5000 patients attending our hospital during two epidemic waves, dominated by the Delta and Omicron variants, respectively. Moreover, we compared the viral loads present in the nasal nostrils and in the anterior oral cavity of Omicron-infected patients in order to assess whether nasal swab collection, which is generally the preferred practice for ADT testing, could still be suitable for Omicron descending variants, or whether it should be switched to a mouth swab, a sample type that could also reduce patient discomfort. Finally, an *in silico* study was performed to evaluate the effect of mutations on the conformation of the N protein in the Omicron and Delta variants and their possible impact on molecular recognition by ADTs.

## 2. Materials and Methods

This paper refers to the STARD 2015 guidelines [27] for the evaluation and reporting of diagnostic test accuracy.

### 2.1. Study Population

The test performance assessment included samples from 5175 subjects, either symptomatic or asymptomatic, who were referred to the IRCCS Sacro Cuore Don Calabria Hospital (Italy) between 1 October 2021 and 15 July 2022 for SARS-CoV-2 testing, most of whom were tested prior to hospital procedures or were contacts of infected persons. The enrolled subjects were assigned to either the Delta (1 October 2021 to 15 January 2022; n = 2726) or Omicron (from 16 January 2022 to 15 July 2022; n = 2449) wave, according to the viral variant dominating in the Veneto Region in the corresponding period. No information on the presence of symptoms was available. The inclusion criterion was the availability of results from both a SARS-CoV-2 ADT and RT-PCR on two parallel samples collected on the same day. According to the hospital’s procedures, for each person, two different nasal/nasopharyngeal swab samples were concomitantly collected by trained healthcare personnel, one for ADT, according to the manufacturer’s instructions, and the other for routine SARS-CoV-2 RT-PCR using eSwab^®^ (COPAN Diagnostics Inc., Murrieta, CA, USA). Both samples were processed in the laboratory of the IRCCS Sacro Cuore Don Calabria Hospital within two hours of sample collection. Data were retrieved from the database of the internal Laboratory Information Management System (LIMS), including the date of collection, study patient code, age, sex, type of test assay, and result for SARS-CoV-2 testing.

For the comparison of the nose vs. mouth swabs, 61 subjects verified as positive for SARS-CoV-2, according to either a molecular or antigen test, were recruited during the Omicron period. For each subject, one mouth (buccal, internal cheeks, MS) and one nasal (anterior nares, NS) swab were collected in parallel by healthcare staff with eSwab^®^ (Copan, Brescia, Italy). Both samples were analysed by RT-PCR for SARS-CoV-2 detection.

### 2.2. Ethics

The study was conducted in accordance with the ethical principles of the Declaration of Helsinki. Subjects or their legal representatives provided written informed consent. The study was approved by the local Ethics Committee (Comitato Etico per la Sperimentazione Clinica delle Province di Verona e Rovigo), protocol n° 17058/2022.

### 2.3. SARS-CoV-2 Antigen Diagnostic Tests

During the study, different ADTs were used for diagnostic purposes. Their main characteristics are summarized in Table 1. Each ADT test was applied by a nasal or nasopharyngeal swab according to the manufacturer’s instructions, as indicated in Table 1. For the comparison among the different types of assays, the 6 tests used were grouped as follows:Group 1 ADT: Lateral Flow Immunochromatography rapid assayGroup 2 ADT: Microfluidic-based rapid assayGroup 3 ADT: Chemiluminescence-based assay

All the different ADTs mentioned were used without preference during the study period, according to the working needs of the laboratory.

### 2.4. SARS-CoV-2 RT-PCR Analysis

The swab specimens were analysed by routine SARS-CoV-2 RT-PCR. Briefly, RNA was extracted from 200µL of eSwabs medium using the automated Microlab Nimbus workstation (Hamilton, Reno, NV, USA) coupled to a Kingfisher Presto system (Thermo Fisher Scientific, Waltham, MA, USA) or using the EZ1 Advanced XL instrument with EZ1 DSP Virus Kit (Qiagen, Hilden, Germany) according to the manufacturer’s instructions.

RT-PCR was performed using the Bosphore SARS-CoV-2/Flu/RSV IVD panel (Anatolia geneworks, Sultanbeyli/İstanbul, Turkey), targeting the Orf 1a/b and N genes, using a CFX96 Touch Real-Time PCR Detection System (Bio-Rad Laboratories S.r.l., Segrate/Mi, Italy). The amplification cycle threshold (Ct) was determined using CFX Maestro (Bio-Rad). Alternatively, the Real-Time PCR SARS-CoV-2 Panel Kit using NeuMoDx istrument (Qiagen Italia, Milan, Italy) was employed, targeting the N and Nsp2 genes. The Ct value for the N target was used as a proxy of the viral load in the corresponding sample. Cellular RnaseP mRNA was used as am endogenous control for the RT-PCR.

### 2.5. SARS-CoV-2 Genome Sequencing

Genomic sequencing for SARS-CoV-2 variant or lineage identification was applied to RT-PCR-positive samples from 168 patients from the Delta and Omicron waves. Reverse-transcription was performed with the SuperScript™ VILO™ Master Mix (Thermo Fisher Scientific, Waltham, MA, USA) in 20 μL of reaction volume, as per the user manual. The SARS-CoV-2 genome was amplified, according to the manufacturer’s instructions, with the Ion AmpliSeq™ SARS-CoV-2 Insight Research Panel (Thermo Fisher Scientific, Waltham, MA, USA), with two primer pools protocol covering the whole SARS-CoV-2 genome. Amplified fragments were used to prepare barcoded libraries for massive parallel sequencing using the Ion AmpliSeq™ Library Kit Plus (Thermo Fisher Scientific, MA, USA), as reported in the user guide. The barcoded libraries were purified with Agencourt™ AMPure™ XP Reagent (Beckman Coulter), eluted in 50 µL of TE buffer, analysed on the 4150 TapeStation System (Agilent, Santa Clara, CA, USA) (average size 250–400 bp), and quantified by a Qubit™ Fluorometer (Thermo Fisher Scientific, Waltham, MA, USA). Each of the prepared libraries was diluted to 100 pM and pooled together; 30 pM of the library’s pool was loaded on the Ion Chef™ Instrument (Thermo Fisher Scientific, Waltham, MA, USA) for clonal amplification and chip loading. The clonally amplified libraries were, shortly afterwards, subjected to next-generation sequencing on the Ion GeneStudio™ S5 System (Thermo Fisher Scientific, Waltham, MA, USA), on the Ion 520 or 530 chips.

### 2.6. Bioinformatic Analysis of Genome Sequences

The sequencing results were analysed in the Torrent Suite™ Software (v 5.14.1) using the SARS-CoV-2 plugins [i.e., generateConsensus, SARS-CoV_2_annotateSnpEff, SARS-CoV_2_variantCaller, SARS-CoV_2_coverageAnalysis (Thermo Fisher Scientific, Waltham, MA, USA)] with standard configuration. BAM files were visualized in the Integrative Genomic Viewer (IGV). FASTA consensus files were used for a lineage analysis with the Pangolin COVID-19 Lineage Assigner https://pangolin.cog-uk.io (accessed on 30 December 2023); sequences that passed the QC during Nextclade v2.8.1 analysis https://clades.nextstrain.org/ (accessed on 30 December 2023) were further submitted to GISAID https://gisaid.org/ (accessed on 30 December 2023). Specific sample mutations of the N gene were obtained from the CoV-GLUE database http://cov-glue.cvr.gla.ac.uk/#/home (accessed on 30 December 2023), and their frequency of occurrence was determined.

### 2.7. Nucleocapsid (N) Protein Mutation Analysis

The N-terminal domain (NTD) and C-terminal domain (CTD) X-ray structures were retrieved from the Protein Data Bank (PDB), with the accession IDs 6VYO and 6WZO, respectively [4]. The mutations in the proteins were mapped using the PyMOL software (v2.4.1) [28]. For the intermediate linker region (LKR), we used AlphaFold2 [29] for modelling the full-length protein (Appendix A). After the mutation mapping, we ran the InterfaceResidues.py script to identify if they belonged to the dimerization interface. Using the Mutagenesis Wizard function in PyMOL, we changed residue S310 from Serine to Cysteine (S310C) (Appendix A). The rotamer chosen for the substitution also did not cause any conflicts. This process was repeated for both chains in the dimer. We estimated the variation in protein folding free energy (ΔΔG) brought on by mutations, carrying out a qualitative evaluation of the N protein stability by using the webservers DynaMut [30] and DynaMut2 [31]. Additionally, using the DynaMut2 tool to compute the changes in vibrational entropy (ΔΔSVibENCoM), we investigated the potential impact on the monomer’s flexibility. Due to the uncertainties in the modelling of the intrinsically disordered regions present in the LKR, this analysis was performed on the solved domains of the protein, i.e., the RNA-binding domain (i.e., NTD) and the N dimerization domain (i.e., CTD) of the N protein structures (PDB IDs 6wzo and 6vyo, respectively).

### 2.8. Statistical Analysis

Continuous variables were summarized with means, standard deviations (SD), and ranges (confidence interval, CI), while count variables were summarized with absolute and percentage frequencies. The normality distribution of the data was assessed using the Shapiro–Wilk test. A comparison of the N gene Ct values between groups was performed using the Wilcoxon test. A comparison of the sensitivity and specificity of ADTs between the Delta and Omicron periods was performed using the two-sample Z-test for proportions. A comparison of the Ct values across time and sampling sites was performed, stratifying the analyses accordingly. R v. 4.2.3 [32], Graphpad Prism v. 10.1.0(316) (GraphPad Software, Boston, MA, USA) and SAS (SAS 9.4 Software, USA) were used to perform statistical analyses.

## 3. Results

### 3.1. Evaluation of ADT Performance in Delta versus Omicron VOCs Period

In order to analyse the performance of the ADTs during the Delta and Omicron waves, we retrospectively evaluated data collected from 5175 patients subjected to both ADTs and RT-PCR tests for SARS-CoV-2 infection in the period from 1 October 2021 to 15 July 2022. The demographic characteristics of the patients are summarized in Table 2. According to the data on the prevalence of viral variants in our region (Veneto, Italy) [33], we divided our study into two periods: the first, from 1 October 2021 to 15 January 2022, when Delta was predominant (Delta wave), and the second, from 16 January 2022 to 15 July 2022, when Omicron was predominant (Omicron wave).

Taking into account the samples with an RT-PCR positive result, we evaluated the Ct values of the N gene and compared the results from ADT positive (+) and negative (−) specimens in the two periods. As expected, significant differences in the Ct values were observed between the ADT+ and ADT− samples, with a significantly lower median Ct value in the first group for both periods (*p* < 0.0001, for both Delta and Omicron periods, Figure 1a,b). When comparing the two waves, significant differences were observed in the median Ct values detected for RT-PCR+/RADT+ (*p* = 0.0009) between the Delta and Omicron periods, indicating a generally higher viral load during the Omicron wave, detected at the RNA level for samples with a positive ADT, whereas no differences were found for RT-PCR+/RADT− (Figure 1c).

We then compared the diagnostic performances of the ADTs between the two SARS-CoV-2 variant periods, using RT-PCR as a reference, and the results are reported in Table 3. During the Delta wave, 122 out of 2726 swabs (4.4%) tested positive by ADT and RT-PCR, and 2512 (92.1%) tested negative by both assays (overall concordance: 96.6%). We found 92 discordant samples (3.4%): 70 (2.5%) that tested negative by ADT and positive by RT-PCR, whereas 22 samples (0.8%) tested positive by ADT and negative by RT-PCR (Table 3). The sensitivity and specificity of ADTs during the Delta wave were 64% (95% CI, 56 to 70) and 99% (95% CI, 99 to 99), respectively (Table 3).

Throughout the Omicron wave, 65 out of 2449 swabs (2.6%) were positive by ADT and RT-PCR, and 2253 (91.9%) tested negative by both assays (overall concordance: 94.6%). We found 131 discordant samples (5.3%), of which 130 (5.3%) tested negative by ADT and positive by RT-PCR, and 1 positive by ADT and negative by RT-PCR. ADTs carried out during the Omicron wave achieved an overall sensitivity and specificity of 33.3% (95% CI, 26.8 to 44) and 100% (95% CI, 99.8 to 100), respectively (Table 3).

For both sensitivity and specificity, the differences between the overall performances of the two tests during the Omicron and Delta periods were found to be statistically significant (both *p*-value < 0.001).The Positive and Negative Predictive values (PPVs and NPVs) of the ADTs for the two variants were calculated, considering the prevalence of SARS-CoV-2 infections during the two waves according to the GIMBE foundation, Italy, “https://www.gimbe.org/ (accessed on 30 December 2023)”, i.e., 2.6% during the Delta and 2.2% during the Omicron wave. The PPVs and NPVs resulted in being 65% (95% CI, 57 to 73) and 99% (95% CI, 98 to 99) for the Delta wave and 94% (95% CI, 85 to 98) and 98% (95% CI, 97 to 98) for the Omicron wave, respectively.

We compared the sensitivity and specificity of the Delta and Omicron periods in more detail by stratifying the different rapid antigen tests according to three types of assay. Specifically, Group 1 included all ADTs based on lateral flow immunochromatography rapid assays, Group 2 referred to microfluidic-based ADTs, and Group 3 included all chemiluminescence-based ADTs. As shown in Table 3, the results showed reduced sensitivities for each group of ADT during the Omicron period. Due to the small number of samples in split groups, statistical significance was only achieved for the overall analysis.

### 3.2. Evaluation of Nucleocapsid Protein Mutations in Delta and Omicron Variants

In order to assess whether specific mutations in the Delta and/or Omicron variants may affect the structure and function of the N protein, we analysed the amino acid sequence variations translated from the SARS-CoV-2 whole genome data, available from positive swabs in our study. Data were collected from 168 patients at the IRCCS Sacro Cuore Don Calabria Hospital in both the Delta and Omicron waves and were submitted to GISAID database. The protein is structured into three principal regions crucial for its activity: an N-terminal domain (NTD) responsible for RNA binding, a C-terminal domain (CTD) involved in dimerization, and an intermediate linker region (LKR) with a serine- and arginine-rich (SR-rich) motif [34], which, when phosphorylated, can regulate discontinuous transcription during the early stages of replication [35].

In the analysed sequences, a total of 33 different mutations were detected in the N protein sequence of the Delta and Omicron samples with respect to the original Wuhan sequence, and the LKR turned out to be the most affected region (Figure 2a). The frequency of each mutation is shown in Figure 2b. We observed that both Delta and Omicron VOCs showed exclusive mutations, e.g., D343G, P80R, and others among the most frequent ones are exclusive to the Omicron variant. Although the structure of the NTD and CTD domains of the N protein were solved, the full-length structure remains difficult to obtain due to protein stability issues and the presence of intrinsically disordered regions (IDRs) [36].

We performed *in silico* modelling of the full-length protein (Appendix A) to evaluate the structural locations of mutations in the Delta and Omicron variants. Due to the uncertainties in the modelling of the IDRs, we focused on the experimentally solved 3D structures of the NTD and CTD (Figure 3a and Figure 3b, respectively). We predicted the differences in folding free energy (ΔΔG) and vibrational entropy (ΔΔSVibENCoM) between the wild type and mutants in order to better understand how mutations may affect the protein stability (Appendix A). A positive ΔΔG indicates an increased stability, whereas a negative ΔΔG indicates a decreased stability. A negative ΔΔSVibENCoM indicates an increase in protein rigidification, while a positive ΔΔSVibENCoM implies an increase in protein structure flexibility. An increase in terms of protein folding energy was predicted for P80R and H300Y and a decrease for D343G and S310C. The D343G mutation was shown to have the most negative ΔΔG, while P80R had the highest positive ΔΔG. These two mutations, both exclusive to the Omicron VOC, were connected with the largest increase (D343G) and decrease (P80R) in vibrational entropy, indicating an effect of these mutations on the Omicron VOC’s structural/dynamic properties, which could lead to different recognitions by antibody-based detection systems.

### 3.3. Mouth versus Nose Viral Load in Omicron-Infected Patients

In order to assess the SARS-CoV-2 viral load at different sites (nose and mouth) during the Omicron wave, a total of 61 symptomatic patients (30 female, 31 male; mean ages of 43 and 44 years, respectively) were tested by RT-PCR in both the mouth and nose. Fifty-one subjects reported mild symptoms. In 49 out of the 61 patients (80% of the total population), the samples were collected less than 4 days after the infection diagnosis, and in 27 (44%) of these, the samples were collected at the onset of symptoms or the following day. Fifty-seven out of the sixty-one patients were positive for at least one of the two swabs (nasal or oral). In particular, 43 of them were RT-PCR positive on both sites, 12 patients were positive only in the nose, and 2 only in the mouth.

Four patients resulted in being negative in both sites, but all of them were sampled more than 6 days after symptoms onset. So, the number of RT-PCR-positive NS was higher than that of positive BS. In line with these results, when analysing the samples’ Ct, we found that the nose site presented lower Ct values, corresponding to a higher viral load compared to the mouth (Wilcoxon test, *p* < 0.001) (Figure 4a).

Possible changes in the viral load during the year were also investigated. We found that the nose was the site where the virus was more likely to be detected [significant results for the months of March (*p* = 0.004) and April (*p* < 0.001), Figure 4b]. When performing a breakdown of the data according to days after symptoms onset, a higher viral load was always detected in the nose with respect to the mouth [significance at 1 day (*p* = 0.006), 2 days (*p* = 0.016), 4 days or more (*p* = 0.002), Figure 4c]. After day 5 from symptoms onset, few samples showed a positive signal. Moreover, we analysed the Ct trend after symptoms onset at the two sampling sites based on the different identified Omicron subvariants (AY.4, BA.1, BA.1.1, BA.2, BA.2.9, BA.2.18, BA.5.1, and BA.5.2, with BA.2 being the most frequent). Appendix A shows that none of the subvariants showed a higher presence in one of the two collections sites.

## 4. Discussion

The genome of Omicron subvariants contains more than 50 mutations [37], many of which have been associated with an increased transmissibility, variable disease severity, and the potential to evade immune responses acquired after SARS-CoV-2 vaccination or infection with a previous variant. Few studies have attempted to investigate the impact of mutations in the N protein on the diagnostic performance of ADTs, with conflicting results [8,18,19,20,21,22,38]. Due to a possible change in the tropism, it has been suggested that the detection of the Omicron variant could be favoured in oral swabs compared to nasal swabs [24]. In the present study, we monitored the performance of the ADTs in a real-world scenario, studying a cohort of more than 5000 subjects across the Delta and Omicron waves. We also assessed the viral load of Omicron at different sites (specifically, the nose and mouth). Moreover, an *in silico* study at the amino acid level was performed to investigate the possible effect of mutations on conformational changes in the Omicron and Delta variants’ N protein, which may affect its recognition by antigen tests.

Our results indicate that, as expected, for both the Delta and Omicron waves, the ADTs could only detect samples with a relatively higher viral load compared to the molecular test based on RT-PCR (Figure 1a and Figure 1b). All the used ADTs targeted the N protein, and the Ct values analysis was performed focusing on the N gene. Significant differences were observed in the Ct values detected for antigenic-positive samples (RT-PCR+/RADT+) between the Delta and Omicron periods, indicating even a higher viral loads at the RNA level in ADT-positive samples for the Omicron period compared to the Delta period (Figure 1c). No differences were found in the viral load of the antigenic-negative samples (RT-PCR+/RADT−).

Importantly, when evaluating the diagnostic performance of the ADTs between the two SARS-CoV-2 variant periods (Table 3), the ADTs showed a decrease of about 30% in sensitivity during the Omicron compared to the Delta period, accompanied by a slight but significant increase in specificity (Table 3). As the Ct values of ADT-negative samples were similar in the Delta and Omicron waves and ADT-positive samples presented even lower Ct in Omicron compared to Delta, we can conclude that the decrease in ADT sensitivity for Omicron was not due to a lower viral load, but was more likely due to a change in the N protein. A possible theory is that this reduced sensitivity was due to a reduced recognition of N antigen by the ADT; this hypothesis is supported by the analysis of the N protein structure *in silico*. In fact, to investigate the possible effect of mutations present in the N protein on the ADTs’ performances [39], we evaluated the SARS-CoV-2 whole genome data obtained from positive swabs of 168 patients from both the Delta and Omicron waves. Focusing the mutation analysis on the N protein amino acid sequence, i.e., the target of ADTs, we observed mutations localized in the NTD, the LKR, and the CTD (Figure 2). Our results confirmed the literature data, showing that mutations in the SARS-CoV-2 N protein mainly accumulate within intrinsically disordered regions, probably due to the functional importance of the NTD and CTD [34,36,40]. In the Omicron LKR region, the co-occurring mutations R203K and G204R are the most common mutations, with a frequency of >60% across all sequences [36,37,38]. Within the NTD and CTD domains, the folding free energy and vibrational entropy analysis indicated that P80R and D343G, both exclusively present in the Omicron variant, were shown to putatively alter the dynamic properties of the protein (Appendix A), strongly suggesting that these mutations may affect the N-protein stability and dynamicity and reduce the performance of antigenic assays. Moreover, from the literature data, the P13L and E378Q mutations also present in the N and C arms of Omicron variants, respectively, were predicted to destabilize the N protein [39].

An additional hypothesis that could explain the observed variations in the ADT sensitivity is the different amounts of nucleocapsid protein that could be shed during infections with different virus variants. Rao et al. showed that Omicron samples had lower ratios of antigen to RNA compared to Delta, which leads to a possible explanation for this result when using Ct values as a reference [20].

To establish whether a reduced sensitivity of ADTs could be due to a shift in viral tropism with a preferential location in the mouth compared to the nose, we evaluated the viral load in nasal nostrils and buccal swabs in Omicron-infected patients. The choice of NS and MS was performed in order to reduce the patients’ discomfort. For this purpose, 61 patients from the Omicron wave underwent RT-PCR testing on swabs collected from both sites. We found that the nasal site had significantly lower Ct values and, therefore, a higher viral load than the oral site. Furthermore, when the viral load was examined according to the different periods in which SARS-CoV-2 Omicron subvariants were prevalent, the nose was always confirmed as the sample type in which the virus was more detectable, especially during the first 3 days after symptoms onset (Figure 3). Molecular characterization showed that the preferred virus localization in the nose vs mouth was independent from the specific Omicron subvariant.

Overall, our results indicate that the nose is the best sampling site to maximize virus detection for a diagnosis of Omicron infection. Nasal mid-turbinate swabs can be used for both ADTs and molecular assays to provide safe and reliable results as an alternative to nasopharyngeal swabs, in an effort to reduce patient discomfort [41,42].

This study has several limitations. First, the lack of clinical characteristics of the patients included in the comparison of ADTs’ sensitivity in Omicron vs. Delta infections. In fact, previous studies have shown a very low sensitivity of rapid antigen tests in asymptomatic patients, and only a moderate decrease in sensitivity for symptomatic Omicron infections [20,21,22,38,43]. Second, 6.8% of the analysed samples derived from multiple hospital accesses were from individuals who participated more than once in the study, so this could be a possible confounder. Third, the use of different ADTs throughout the study may have introduced some bias into the analysis, due to possible heterogeneous results from the different ADTs. However, the stratification analysis according to the type of assay and the high number of participants provided confidence in the reliability of the results and their interpretation, indicating a substantially reduced sensitivity of ADTs for Omicron infections.

## 5. Conclusions

In conclusion, real-life data from a large number of subjects strongly support the evidence of a substantially reduced detection rate of Omicron infections by ADTs, confirming and extending circumstantial evidence from previous studies.

This drop in sensitivity should be taken into consideration in establishing testing strategies and monitoring infection prevalence. The emergence of new variants, as well as new mutations affecting the N protein structure, might further affect ADTs’ diagnostic performance, which could require assay revalidation to maintain efficient and reliable screening and diagnostic strategy programs. Our study suggests that ADTs should be adapted to better detect Omicron-descending variants.

## Figures and Tables

**Figure 1 viruses-16-00657-f001:**
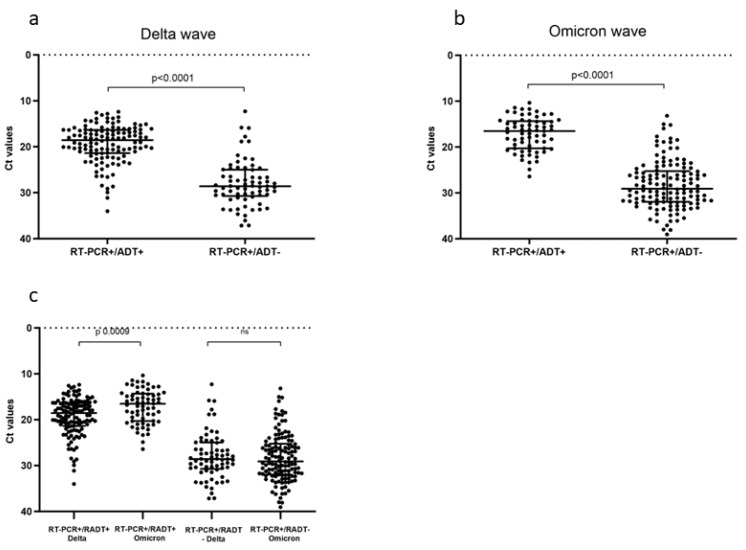
Comparison of Ct values of SARS-CoV-2 RT-PCR according to ADT results. (**a**) Panel shows results during Delta and (**b**) panel during Omicron waves, respectively. (**c**) Comparison of Ct values of ADT-positive and -negative results for Delta and Omicron waves. Each dot plot represents an individual Ct value, error bars represent median with interquartile range (IQR). Wilcoxon test was applied to compare the difference of Ct values between the two groups. *p* < 0.05 was accepted as significant difference.

**Figure 2 viruses-16-00657-f002:**
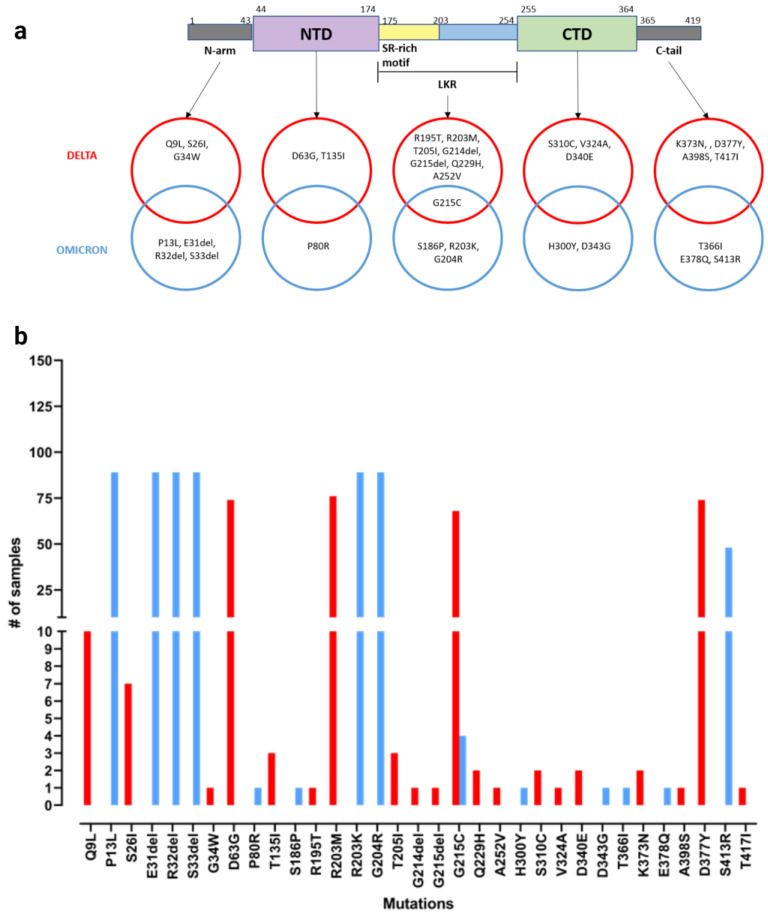
Mutation in N protein of Delta and Omicron sequences identified in infected subjects included in the study. (**a**) Panel represents the modular structure of the SARS-CoV-2 N protein with mutations identified for Delta (in red) and Omicron (in blue) variants. (**b**) Panel shows mutation frequencies in Delta (in red) and Omicron (in blue) variants.

**Figure 3 viruses-16-00657-f003:**
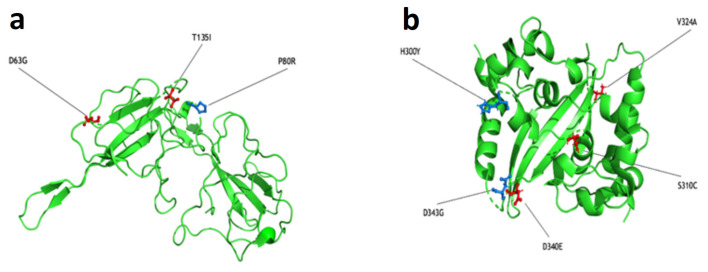
Spatial representation of Delta and Omicron mutations within the N protein. The 3D structure of the NTD (**a**) and CTD (**b**) are reported. NTD and CTD structures are associated with the PDB IDs 6vyo and 6wzo, respectively. Mutations in red are characteristic of the Delta variant, whereas the blue ones belong to the Omicron variant.

**Figure 4 viruses-16-00657-f004:**
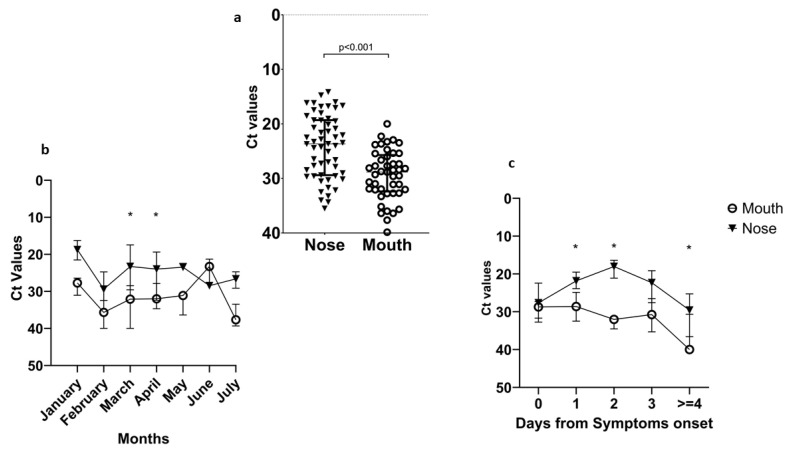
Viral load during the Omicron wave in different upper respiratory tract sampling sites. (**a**) Shows the Ct values detected in the nose (left) and in the mouth (right). Viral load dynamics in mouth and nose. (**b**) Shows the median Ct values and the IQR detected in the mouth (triangle) and in the nose (circle) across the year. (**c**) Shows the median Ct values and the IQR detected in the mouth (circle) and in the nose (triangle) based on days after symptoms.

**Table 1 viruses-16-00657-t001:** Characteristics of SARS-CoV-2 ADTs used in this study. Sensitivity and specificity as reported by test manufacturers [ECDC, COVID-19 In Vitro Diagnostic Devices and Test Methods Database, available at https://covid-19-diagnostics.jrc.ec.europa.eu/devices/, (accessed on 30 December 2023)].

ADT Group	Commercial Name	Manufacture	Assay Type	Target Protein	Sampling Site	Sensitivity	Specificity
**1**	STANDARD Q COVID-19 Ag Test 2.0	SD Biosensor, Inc. Korea	Lateral Flow, Immuno-chromatography Rapid ADT	N	Nasal swab	94.94%	100%
**1**	Panbio™ COVID-19 Ag Rapid Test Device	Abbott Rapid Diagnostics Jena GmbH, Germany	Lateral Flow, Immuno-chromatography Rapid ADT	N	Nasal swab	98.1%	99.8%
**1**	Green Spring SARS-CoV-2 Antigen Rapid Test Kit	Shenzhen Lvshiyuan Biotechnology Co. Ltd., China	Lateral Flow, Immuno-chromatography Rapid ADT	N	Nasal swab	96.77%	100%
**2**	FREND™ COVID-19 Ag	NanoEntek, South Korea	Microfluidic-based rapid ADT	N	Nasophar. Swab	94.12%	94.12%
**3**	MAGLUMI^®^ SARS-CoV-2 Ag	Shenzhen New Industries Biomedical Engineering Co., Ltd. China	Laboratory-chemiluminescence-based ADT	N	Nasophar. Swab	97.7%	99.6%
**3**	LIAISON^®^ SARS-CoV-2 Ag	DiaSorin, Inc	Laboratory chemiluminescence-based ADT	N	Nasal swab	99.0%	98.0%

**Table 2 viruses-16-00657-t002:** Descriptive statistics of the ADT study population. A total of 5175 subjects were considered. Subjects were divided according to Delta and Omicron waves.

		Delta Wave		Omicron Wave	
Demographics		Count (n)	Value (%)	Count (n)	Value (%)
Population		2726		2449	
Gender					
	Female	1319	48.38	1216	49.65
	Male	1407	51.61	1233	50.34
**Age (years)**		**Female**	**Male**	**Female**	**Male**
	Lower 95% CI	46.71	46.81	39.01	41.20
	Upper 95% CI	49.68	49.71	42.02	44.33
	Median	48.00	52.00	37.00	41.00

**Table 3 viruses-16-00657-t003:** SARS-CoV-2 ADT results during Delta and Omicron waves. RT-PCR results have been used as reference for calculation of sensitivity and specificity of ADT. Results were also stratified based on the assay type of the test as follow: Group 1 (Lateral Flow Immuno-chromatography rapid assay), Group 2 (Microfluidic-based rapid assay), and Group 3 (Chemiluminescence-based assay). SE: sensitivity, SP: specificity, TN: true negative, TP: true positive, FN: false negative, FP: false positive, and CI: confidence interval.

	Delta		Omicron			Delta		Omicron		
	TP/(TP + FN)	SE (95% CI)	TP/(TP + FN)	SE (95% CI)	*p*	TN/(TN + FP)	SP (95% CI)	TN/(TN + FP)	SP (95% CI)	*p*
**Overall**	122/192	63.5 (56.3, 74.0)	65/195	33.3 (26.8, 44.0)	<0.001	2512/2534	99.1 (98.7, 99.5)	2253/2254	99.9 (99.8, 100)	<0.001
**Group 1**	27/54	50 (36.1, 63.9)	32/96	33.3 (24.0, 43.7)	0.045	549/550	99.8 (99.0, 100)	1496/1497	99.9 (99.6, 100)	0.460
**Group 2**	73/112	65.2 (55.6, 73.9)	22/62	35.5 (23.7, 48.7)	<0.001	1822/1838	99.1 (98.6, 99.5)	614/614	100 (99.4, 100)	0.020
**Group 3**	22/26	84.6 (65.1, 95.6)	11/37	29.7 (15.9, 47.0)	<0.001	141/146	96.6 (92.2, 98.9)	143/143	100 (97.5, 100)	0.026

## Data Availability

The data that support the findings of this study are available in Zenodo repository at the following link DOI 10.5281/zenodo.10640951.

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
