# Peer review of "Wide Real-Life Data Support Reduced Sensitivity of Antigen Tests for Omicron SARS-CoV-2 Infections"

_viruses, 2024, doi:10.3390/v16050657_

Round 1
Reviewer 1 Report
Comments and Suggestions for Authors
In their cross-sectional study, Piubelli et al. performed testing for acute SARS-CoV-2 infections among 5,175 individuals administered to a hospital utilizing both RT-PCR and rapid antigen tests. In total, 387 patients tested positive for acute SARS-CoV-2 infections in the RT-PCR assay. The authors’ analysis showed that rapid antigen tests were more likely to be positive at higher viral loads. Compared to the RT-PCR, the diagnostic sensitivity of rapid antigen tests was at 67 % at the time when SARS-CoV-2 VoC delta was most prevalent. Interestingly, between January and July 2022 when VoC omicron was dominating the COVID-19 pandemic, the rapid antigen tests’ sensitivity was significantly lower, i.e. at 33 %. Analyzing the viral loads at the swab sampling sites, the authors conclude that the differences in sensitivity comparing delta and omicron infected individuals are not due to alterations in the abundance of viral RNA. Supported by in silico analysis they hypothesize, however, that mutations in the nucleocapsid antigen may cause the observed decrease in the rapid antigen tests’ sensitivity for detecting omicron infections. Generally, higher viral loads were found in nasal swabs compared to those sampled in the oral cavity.
The study by Piubelli et al. shows, once more, evidence for the low diagnostic sensitivity of rapid antigen tests for the detection of acute SARS-CoV-2 infections, especially in case of infection with VoC omicron. Taken together, this study could be of high interest for the readership of Viruses. In my opinion, however, several concerns need to be addressed before this manuscript is eligible for publication in the journal.
Specific comments:
Please mention in the abstract how many participants, in total, tested positive for an acute SARS-CoV-2 infection using the RT-PCR assay.
Throughout the manuscript statistical analyses were used that are most suitable for normal distributed data. Please test and indicate whether the results of the parameters tested are, indeed, normal distributed in the study cohort. If not, I highly recommend using non-parametric statistical tests to evaluate differences between groups (e.g., Wilcoxon test instead of student’s t-test, and Kruskal-Wallis test instead of ANOVA) as well as depicting medians and interquartile ranges instead of means and standard deviations.
Six different rapid antigen tests were utilized in the study. If possible, please show and discuss the sensitivity and specificity comparing the different rapid antigen tests used in an additional table or graph.
The authors show that the abundance of viral RNA in nasal swaps was often higher than in oral swabs. Therefore, it could be of interest to state in the manuscript whether the different rapid antigen test kits employed relied on oral or nasal sampling when performed according to the manufacturers’ instructions.
Please clarify whether the oral swabs were performed as tongue swabs or as oropharyngeal swabs (including sampling in the upper pharynx of the participants). Oropharyngeal swaps are often considered to have higher diagnostic sensitivity for the detection of SARS-CoV-2.
The authors show that differences in rapid antigen tests’ sensitivity comparing infections with delta and omicron are not due to alterations in the viral loads at the sampling site. Therefore, they suggest that the sequence and/or structure of the nucleocapsid protein in different virus variants itself may influence detection with rapid antigen tests. However, there is also the possibility that different amounts of nucleocapsid are shed during infections with different virus variants independent from the viral load at the sampling site. This could also explain the observed variations in the tests’ sensitivity. Please add a paragraph to the discussion section of the manuscript exploring this thought.
Was it, in theory, possible that certain individuals participated more than once in the study because they were administered to the hospital several times during the study period? If so, please mention this possible confounder in the limitations section of the manuscript.
In my opinion, the results shown in Figure 2b would be easier to read if percentages of samples having a certain mutation were depicted on the y-axis instead of total numbers of samples.
Comments on the Quality of English LanguageSeveral typos and grammatical errors were found while reading the manuscript. I recommend the authors to carefully read the paper and correct the errors.
Reviewer 2 Report
Comments and Suggestions for Authors
This article attempts to study in depth the cause of the low sensitivity of antigen tests specifically with the omicron variant in relation to the previous variant, delta.
It is an interesting article and the methodology is well designed and carried out. In addition, the sample size is adequate.
However, in my opinion, there are some issues that should be clarified and improved for it to be accepted for publication.
In the material and methods section for the comparative study of the antigen test against the different variants and for comparison with real-time PCR, the sample collected is said to be a nasopharyngeal swab.
However, for the comparison study of sample collection location sites, nasal and buccal swabs were collected. Does oral exudate in this case refer to pharyngeal swab or only to exudate from the mouth surface?
If it was buccal, why was the difference with a pharyngeal exudate not studied? This would significantly vary the results of the viral load, in addition to being more useful to propose a sampling location easier to collect. Taking into account that the sample proposed for collection is a nasopharyngeal swab the study would have been better considering nasal or pharyngeal samples instead of mouth.
As for the antigen tests used, they already say that it is a limitation of the study that there are several different antigens test , but considering the pandemic situation it is the usual. However, due to the differences in sensitivity and specificity of these techniques, perhaps they could indicate the number of samples that were made for each test.
About the influence of mutations in the capsid protein, in the discussion they talk about the R203K and G204R mutations which are the most frequent in LKR region, how did these mutations affect the protein in the study? or were they not studied? and in this case, why?
Another issue to be improved is the tables and figures:
Table 1: Improving the header, maybe add the number of samples in each test
Table 2: Improving data exposure
Table 3: there should be two different tables, one for each variant and in my opinion the positive and negative data would be superfluous.
Figure 2. I would divide it in 2 different ones, one with panels a and b, and the other with panels c and d.
Table 4: the descriptive data part should be improved, perhaps even written in the text. I do not understand the importance of data such as vaccination status, lineage type or the presence or absence of symptoms, since they are not subsequently used for the analysis of the results. Part B, should be deleted and the results on concordance, etc. should be written in the text.
Despite these suggestions, if the modifications are made, especially to the tables and figures, I think the article is interesting and highlights something that was experienced in diagnostic laboratories throughout the pandemic.

Reviewer 3 Report
Comments and Suggestions for Authors
The experiments described in the manuscript suggest a reduction in the sensitivity of antigen diagnostic tests (ADTs) depending on the identified SARS-CoV-2 variant. The Authors showed that the sensitivity of ADTs was twice as high for the Delta as for Omicron variant. In addition, by sequencing the SARS-CoV-2 genomes, the Authors speculate what mutations (in what parts of the N gene) might affect the decrease in sensitivity of ADTs.
In the Reviewer's opinion, the manuscript will be able to be published after the Authors have made corrections according to the Reviewers' suggestions.
Major comments:
1. In the Reviewer's opinion, it will be more important for the common reader to show the number of assays used with a specific product, as well as to demonstrate the comparison of sensitivity and specificity in the Authors' studies with these parameters indicated by the manufacturers, than the data presented in Table 1 from the manufacturer's brochure.
I suggest that from Table 1 the data on "Manufacture", "Assay type", "Target Protein" once "Sampling Site" should be moved to the text, and in their replacement should be placed values indicating the number of determinations using a specific assay with the values for sensitivity and specificity achieved in the Authors' studies.
2. The Reviewer does not agree with the authors' opinion expressed in lines 391-393: "reduced sensitivity of ADT could be due to a shift in viral tropism with preferential location in an upper respiratory tract site, we compared the viral load in nose and mouth". From what basis did the Authors make such a deduction? Alternatively, this conclusion could have been made if the Authors had examined nasopharyngeal swabs for these 61 individuals. The fact that the Authors detected Omicron sub-variants in the nose and mouth does not prove that they were not present in the nasopharynx.
3. The reviewer understands that the authors wanted to enrich the study with prospective samples. But why didn't they collect the swabs in the same way as the retrospective samples? Then the 61 samples would have blended into the overall study. And yes, for the two reasons described above, none of them fit into the larger retrospective sample pool, nor can the tropism of the virus to specific cells be inferred. The Reviewer asks for a reason for the inclusion of these 61 samples in the study, as the reason indicated by the authors is an erroneous assumption of the experiment.
Minor comments:
1. References should be in square brackets.
2. Adjust the text in Results and Discussion.
3. Increase the font in Tables.
Round 2
Reviewer 3 Report
Comments and Suggestions for Authors
The manuscript can be published in its present form.